# The master cell cycle regulator APC-Cdc20 regulates ciliary length and disassembly of the primary cilium

Weiping Wang, Tao Wu, Marc W Kirschner*

Department of Systems Biology, Harvard Medical School, Boston, United States

**Abstract** The primary cilium has an important role in signaling; defects in structure are associated with a variety of human diseases. Much of the most basic biology of this organelle is poorly understood, even basic mechanisms, such as control of growth and resorption. We show that the activity of the anaphase-promoting complex (APC), an E3 that regulates the onset of anaphase, destabilizes axonemal microtubules in the primary cilium. Furthermore, the metaphase APC co-activator, Cdc20, is specifically recruited to the basal body of primary cilia. Inhibition of APC-Cdc20 activity increases the ciliary length, while overexpression of Cdc20 suppresses cilium formation. APC-Cdc20 activity is required for the timely resorption of the cilium after serum stimulation. In addition, APC regulates the stability of axonemal microtubules through targeting Nek1, the ciliary kinase, for proteolysis. These data demonstrate a novel function of APC beyond cell cycle control and implicate critical role of ubiquitin-mediated proteolysis in ciliary disassembly.

## Introduction

The primary cilium is a non-motile microtubule-based organelle, organized by the mother centriole (basal body). Though known for over a century, it received little interest until it was appreciated that it is highly enriched in receptors for various signaling pathways, including PDGF, calcium, receptor tyrosine kinases, Wnt, and Hh (*Satir et al., 2010*). The assembly of primary cilium is coupled to cell cycle exit and entry into quiescence. The primary cilium functions as a tumor suppressor organelle that regulates cell proliferation and differentiation (*Pan et al., 2012*). Dysfunctions of the primary cilium cause a set of human diseases, classified as ciliopathies, such as polycystic kidney diseases (*Hildebrandt et al., 2011*). Cancers are also associated with the loss of cilia, due to misregulated cell proliferation through ciliary signaling (*Plotnikova et al., 2008*).

Ciliopathies are highly associated with defective ciliogenesis and abnormal ciliary length. The assembly/disassembly of cilia and maintenance of ciliary length are tightly regulated through control of intraflagellar transport (IFT) and axoneme modification (*Avasthi and Marshall, 2011*). Ciliary length is regulated by the balance of the bidirectional transport of microtubule subunits and associated protein complexes through IFT (*Pedersen and Rosenbaum, 2008*). Importantly, the extension/resorption of primary cilia also involves changes of the stability of axonemal microtubules, which are controlled by post-translational modifications. For example, Aurora kinases stimulate disassembly of cilia and flagella by phosphorylation and activation of tubulin deacetylase (*Pan et al., 2004*; *Pugacheva et al., 2007*). Another key post-translational modification during ciliogenesis is ubiquitylation. In Chlamydomonas flagella, there is an increase in the levels of protein ubiquitylation during flagellar resorption (*Huang et al., 2009*). However, the precise mechanism for how ubiquitylation might control the process of ciliogenesis is not known.

It is curious that many mitotic regulators have been found to be involved in ciliogenesis, such as Aurora A kinase (*Pugacheva et al., 2007*) and NIMA-related kinases (*Moniz et al., 2011*). Recently, it was shown that anaphase-promoting complex (APC), the key ubiquitin E3 ligase that governs cell cycle

*For correspondence: marc@hms.harvard.edu

Competing interests: The authors declare that no competing interests exist.

**eLife digest** The majority of cells in the human body have small hair-like structures that project from the cell surface. These structures, known as primary cilia, are involved in sensing light and touch, and they are also required for an organism to develop normally. Defects in cilia result in a wide range of human diseases that are collectively known as ciliopathies. These include polycystic kidney disease and Bardet–Biedl syndrome. Ciliary disorders can also affect almost every organ in the body leading to blindness, obesity, diabetes, and cancer.

Cilia are dynamic structures that are dis-assembled when cells start to divide and are then re-assembled when cells are quiescent. The anaphase promoting complex (APC) has a critical role during cell division and targets key proteins that need to be degraded at specific times during this process. APC is localized in the basal body, which is found at the bottom of cilia, and it works together with a number of proteins which assist its function.

Wang et al. now report that a complex formed by APC and its co-activator protein Cdc20 has two functions at the basal body: it is needed to maintain the optimal length of the cilia in quiescent cells and to shorten the cilia when cells exit from quiescent stage.

Wang et al. also investigated the role of Nek1, an enzyme that is localised in the basal body. It was found that reducing the level of Nek1 in quiescent cells resulted in the formation of defective cilia, suggesting that this enzyme controls the stability and integrity of cilia. Moreover, when cells undergo division, the APC-Cdc20 complex targets the Nek1 enzyme, causing it to be degraded and allowing the cilia to be disassembled. A detailed understanding of how cells maintain the length of cilia could lead to the development of new approaches for the treatment of human ciliopathies.

progression is localized to the basal body of the motile cilia in multiciliated *Xenopus* epidermal cells, where it has some role in regulating ciliary polarity (*Ganner et al., 2009*). In mitosis, APC sequentially recruits co-activators (Cdc20 followed by Cdh1) and cooperates with two specific E2s (UBCH10 and Ube2S) to control proteolysis of key regulators of the metaphase to anaphase transition (*Fujita et al., 2009*; *Fujita et al., 2009*). APC also has a less well understood role in regulating other cellular processes, such as neurogenesis, metabolism, and myogenesis (*Kim et al., 2009*; *Eguren et al., 2011*).

In mitosis, the activity of APC-Cdc20 is tightly controlled at multiple levels. Regulation of phosphorylation of APC subunits and Cdc20 is essential for the timing of APC activation during mitotic progression (*Kramer et al., 2000*); APC-Cdc20 is also modulated by its inhibitor proteins (*Luo et al., 2000*; *Madgwick et al., 2006*). In addition, protein levels of APC cognate E2s, UBCH10 and Ube2S are crucial for the activation of APC-Cdc20, and play a role in the override of mitotic arrest (*Reddy et al., 2007*; *Garnett et al., 2009*). Although the function and regulation of APC-Cdc20 during mitosis and meiosis is thoroughly studied (*Kramer et al., 2000*; *Madgwick et al., 2006*), the non-mitotic roles of APC-Cdc20 remain largely unknown. It was recently found that APC-Cdc20 is localized to the centrosome of postmitotic neurons and plays an important role in dendrite morphogenesis (*Kim et al., 2009*). It would therefore be of interest to explore the potential activity of Cdc20 in quiescent and differentiated cells. All of this circumstantial evidence suggests that there may be a value in investigating whether APC has a broad regulatory role in the primary cilium. We report here that APC-Cdc20 is central to the regulation of the primary cilium. The APC is localized to the basal body of primary cilia of human epithelial cells, where it negatively controls the length and stability of the axonemal microtubules. During exit from the quiescent (G0) state APC-Cdc20 is activated and drives ciliary resorption.

## Results

### APC-Cdc20 is localized to the basal body of the ciliated human epithelial cell

Given the well-established role for APC in dividing cells, we are interested in whether APC is still active and what role it might have in quiescent ciliated cells. Human hTERT-RPE1 cell line is an established model to study the assembly/disassembly of primary cilia (*Pugacheva et al., 2007*). Greater than 80% of RPE1 cells are ciliated after serum-starvation for 48 hr, and disassembly of primary cilia occurs 1–2 hr after re-adding serum. We found that APC is still active in cytoplasmic extracts of quiescent ciliated

RPE1, using the canonical APC substrate Securin (*Figure 1—figure supplement 1*). Immunostaining showed that APC subunit 2 (APC2) is localized to the basal body of the ciliated cells (*Figure 1A* and *Figure 1—figure supplement 2A*), consistent with the previous report that APC is localized to the basal body of motile cilia in the *Xenopus* epidermis (*Ganner et al., 2009*). Co-localization of APC2 with dynactin subunit P150 (*Guo et al., 2006*) further confirmed its localization to the mother centriole (*Figure 1—figure supplement 2B and 2C*). Furthermore, we found that the APC co-activator Cdc20 is also localized to basal body of the primary cilium (*Figure 1B*), whereas Cdh1 did not show such localization (*Figure1—figure supplement 3*). Interestingly, Cdc20 was not observed at the centro-some of cycling interphase non-ciliated cells (*Figure 1B* and *Figure1—figure supplement 2D*), indicating that Cdc20 is specifically recruited to the basal body during ciliogenesis, rather than generally binding to the interphase centrosome. Western blotting showed that Cdc20 protein levels are dramati-cally reduced after serum starvation for 24 hr, due to the exit of cells from proliferation (*Figure 1—figure supplement 2E*). Importantly, Cdc20 protein levels were significantly elevated at 48 hr after serum starvation, when >80% of the cells are ciliated, consistent with the appearance of this protein at the basal body. Thus, our data suggest that APC-Cdc20 may have a special role at the basal body after initiation of ciliogenesis.

## APC-Cdc20 negatively regulates the length of the primary cilium

We then tested whether Cdc20 is required for some aspect of ciliogenesis. Reduction of APC subunit 2 (APC2) by siRNA significantly increased the length of primary cilia marked by acetylated tubulin staining (*Figure 2A–C*). Consistently, the depletion of Cdc20 by siRNA had no effect on the percentage of ciliated cells, but there was a significant increase in the ciliary length (*Figure 2D–F*). Negative Ki-67 staining demonstrated that these ciliated cells after the knockdown of either APC2 or Cdc20 are still in quiescent stage (*Figure 2—figure supplement 1A*). In contrast, knockdown of Cdh1 by siRNA gave no obvious ciliary phenotype (*Figure 2—figure supplement 2A*). This result is contradictory to a previous study proposing that APC-Cdh1 promote ciliogenesis (*Miyamoto et al., 2011*). Reciprocally, we asked what the effect would be for overexpression of Cdc20. When we transiently transfected cells with plasmid encoding Cdc20-GFP, we found it localized to the centrosome of quiescent cells, and completely suppressed the formation of primary cilia (*Figure 2G,H*). This phenotype was confirmed by Immunostaining of Arl13b, another specific marker for ciliary axoneme (*Figure 2—figure supplement 2B*). These results suggested that Cdc20 negatively controls the length of the primary cilium.

To further validate the effect of Cdc20, we used a small molecule inhibitor of APC, proTAME. Previous studies showed that cell-permeable proTAME efficiently inhibits the APC activity through disruption of the association of APC with its co-activators (*Zeng et al., 2010*; *Zeng and King, 2012*). The advantage of small molecule inhibition over RNAi is that proTAME exerts its inhibitory effect immediately after drug treatment, which helps differentiate the ciliary phenotype from the possible secondary effect caused by impaired cell cycle control. The length of preformed cilia was increased from 4µm to 7µm after treat-ment of fully ciliated quiescent cells with proTAME for 3 hr (*Figure 3*), consistent with the Cdc20 knockdown or APC2 knockdown phenotype. Negative Ki-67 staining shows that the ciliated cells are still in quiescent stage after short-term treatment with this drug (*Figure 2—figure supplement 1B*). This result supports the view that the activity of APC is required for maintaining the proper length of primary cilia.

## APC activity is required for serum-induced resorption of primary cilia

These observations suggested that activation of APC-Cdc20 might be responsible for the destabilization and shortening of ciliary axoneme when quiescent cells re-enter the cell cycle. To test this hypothesis, we treated serum-starved cells with proTAME when adding serum; serum causes ciliary resorption and subsequent re-entry into the cell cycle. Primary cilia on DMSO-treated control cells significantly shortened and resorbed 4 hr after serum addition. ProTAME treatment delayed the shortening and resorption of cilia (*Figure 4A,B*), demonstrating that APC activity is required for efficient ciliary resorption before the exit of cells from quiescent phase. Suppression of either Cdc20 or APC2 by siRNA also significantly delayed ciliary shortening (*Figure 4—figure supplement 1*). Calcium influx is known to be necessary and sufficient for shortening and resorption of the primary cilium (*Besschetnova et al., 2010*; *Plotnikova et al., 2012*), and we found that proTAME treatment also prevented ciliary shortening induced by Calcium ionophore treatment (*Figure 4—figure supplement 2A and 2B*). Based on these results, APC-Cdc20 at the basal body has two related functions: it maintains cilium length of quiescent cells; it is required for shortening and resorption of the cilium.

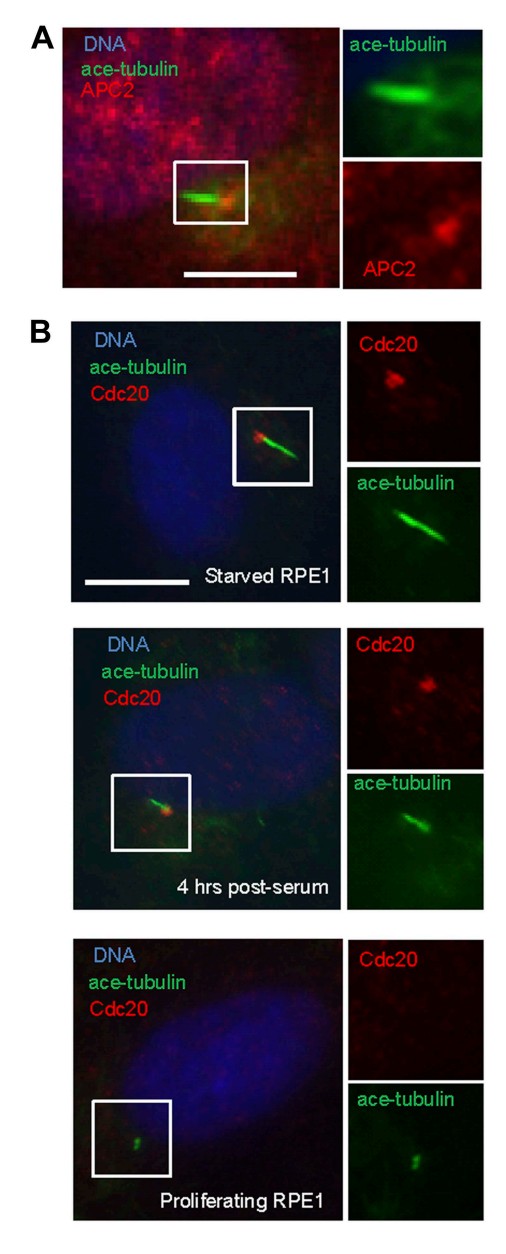

**Figure 1**. APC-Cdc20 is localized to the basal body of the primary cilium. (**A**) Subconfluent hTERT-RPE1 cells were serum starved for 48 hr, fixed, and stained for APC subunit 2 (APC2, red), acetylated tubulin (green) and DNA (blue). (**B**) Proliferating interphase RPE1 cells, serum starved cells, and quiescent cells after serum stimulation were fixed and stained for Cdc20 (red), acetylated tubulin (green), and DNA (blue). Boxes in main images indicate structures shown at higher magnification to right. The scale bars in this figure represent 10 μm.

The following figure supplements are available for figure 1:

**Figure supplement 1**. Degradation of securin in RPE1 cell extracts.

*Figure 1. Continued on next page*

Regulation of APC in mitosis employs a special E2 enzyme, Ube2S, which elongates ubiquitin chains on a substrate through specific linkages at the K-11 position of ubiquitin (*Williamson et al., 2009*; *Wu et al., 2010*). Ube2S keeps APC partially active during mitotic arrest and is required to fully activate APC-Cdc20 (*Garnett et al., 2009*). We were interested in whether Ube2S is also involved in regulation of APC-Cdc20 during ciliogenesis. We found that depletion of Ube2S increased the cilium length in quiescent cells (*Figure 4—figure supplement 3*) and delayed ciliary shortening and resorption upon either serum stimulation or calcium influx (*Figure 4C*; *Figure 4—figure supplement 2C*). These data imply that the cell utilizes the general processes for regulation of APC-Cdc20 activity for both mitosis and ciliogenesis.

## APC-Cdc20 controls the stability of axonemal microtubules through the regulation of Nek1 activity

In mitosis APC ubiquitylates specific regulatory molecules like cyclin B and securin that control the processes of the metaphase–anaphase transition. Similarly, there should be substrates of APC that might regulate the stability of the ciliary axoneme. Aurora-A, a known APC substrate during late mitosis, plays an important role in regulating disassembly of primary cilium (*Pugacheva et al., 2007*). We showed that protein levels of Aurora-A do not decrease upon serum stimulation when APC-Cdc20 is activated (*Figure 5—figure supplement 1A*), consistent with the previous report (*Pugacheva et al., 2007*). This may be explained by previous studies showing that during ciliogenesis, Aurora-A is autophosphorylated at S51 (*Plotnikova et al., 2012*), which protects it from APC-mediated degradation (*Littlepage and Ruderman, 2002*; *Littlepage et al., 2002*). Accordingly, the knock-down of Cdc20 in quiescent cells did not affect Aurora-A protein levels (*Figure 5—figure supplement 1B*). It was indicated by recent studies that APC may regulate ciliogenesis through degradation of dishevelled protein (*Ganner et al., 2009*; *Miyamoto et al., 2011*). However, both specific ubiquitylation assay and in vitro degradation assay showed that dishevelled is not an authentic substrate of APC (*Figure 5—figure supplement 1C and 1D*), contradictory to their argument.

Nek1, the founding member of the NIMA-related kinase family was a good candidate. Genetic mutation in Nek1 was highly associated with short-rib polydactyly syndrome Type Majewski, a new ciliopathy (*Thiel et al., 2011*), and Nek1 is localized to the basal body region (*Shalom et al., 2008*). Both human and mouse fibroblast cells

*Figure 1. Continued*

**Figure supplement 2**. Localization of APC-Cdc20 to the basal body of primary cilia.

**Figure supplement 3**. Cdh1 is not localized to the basal body of primary cilia.

with Nek1 mutations have cilia with abnormal branched structures (*Shalom et al., 2008*; *Thiel et al., 2011*). We found that the depletion of endogenous Nek1 by siRNA in serum-starved RPE1 cells generated cilia with branched structures, revealed by acetylated-tubulin staining (*Figure 5A–C* and *Figure 5—figure supplement 2A*), indicating that Nek1 controls the stability and integrity of axonemal microtubules. Nek1 knockdown in proliferating RPE1 cells did not affect the structure of either cytoplasmic-acetylated tubulin in microtubules or centrosomal gamma-tubulin (*Figure 5—figure supplement 2C*), suggesting a specific and confined role of Nek1 in ciliogenesis. After serum addition, the branched structure can still be efficiently resorbed (*Figure 5—figure supplement 2B*), consistent with the dynamics of ciliary disassembly, indicating that the branched structure is ciliary axoneme. We found that Nek1 levels decreased after serum stimulation (*Figure 5D*), suggesting that the degradation of Nek1 could be essential for triggering the destabilization of the primary cilium. Finally, Nek1 also has a canonical destruction box (RXXLXXXN), the APC recognition motif, right after its coiled-coil domain. We showed by degradation assays in vitro that Nek1 is degraded in HeLa cell extracts, and that this degradation can be suppressed by Emi1, a specific APC inhibitor protein (*Figure 5E*). Nek1 is efficiently ubiquitylated by APC in in vitro ubiquitylation assay involving purified APC (*Figure 5F*). Furthermore, overexpression of Cdc20 leads to reduction of Nek1 levels in quiescent RPE1 cells (*Figure 5G*). These results validate Nek1 as an APC substrate. Moreover, simultaneous depletion of Cdc20 and Nek1 partially rescued the branched cilia phenotype caused by Nek1 knockdown (*Figure 5C*), suggesting that APC-Cdc20 regulates the stability of ciliary axoneme at least in part through antagonizing Nek1 activity.

## Discussion

The assembly, maintenance, and disassembly of the primary cilium are regulated by the control of intraflagellar transport and protein posttranslational modifications (*Besschetnova et al., 2010*). Impairment of ciliary length control and ciliary assembly can lead to inappropriate cell proliferation and defects in signal transduction, causing ciliopathies and perhaps tumorigenesis. Though more and more is understood about which proteins control ciliary function, much of it through human genetics, much less is known about regulatory features that control the timing of ciliogenesis, and with respect to our work here, ciliary resorption. Surprisingly, some of the machinery for regulation is very similar to that used in regulation of the mitotic cycle, which depends significantly on posttranslational modifications. In particular the mitotic process in cells is controlled from the top by the activation of a kinase, Cdk1-cyclin B and an E3 component of the ubiquitin pathway, APC-Cdc20 and APC-Cdh1. We have found that the same Cdc20 is recruited to the basal body of the primary cilium, where in complex with APC it negatively regulates ciliary length, by destabilizing axonemal microtubules. Following serum stimulation of quiescent cells, APC-Cdc20 is fully activated leading to ciliary shortening and resorption.

A previous study in this area claimed that APC-Cdh1 promoted the assembly of primary cilia; a conclusion completely contrary to our results. Although they found, as we did (*Figure 2D–F*), that Cdc20 knockdown promotes ciliogenesis, they explained it by assuming that Cdc20 is an inhibitor of APC-Cdh1, and that Cdc20 needs to be degraded for the activation of APC-Cdh1 to promote the assembly of cilia (*Miyamoto et al., 2011*). They also proposed that APC-Cdh1, but not Cdc20, specifically degrades dishevelled during ciliogenesis. There are several problems with their data and conclusions. We found that Cdc20 is recruited to the basal body of fully formed primary cilia, and there is no evidence for the presence of Cdh1 at the basal body (*Figure 1B* and *Figure 1—figure supplement 3*). Knockdown of Cdh1 does not inhibit the assembly of cilia, contradictory to their model (*Figure 2—figure supplement 2A*). In addition, depletion of APC subunit (APC2) or treatment of ciliated cells with proTAME, which inhibits APC through promoting dissociation of both Cdc20 and Cdh1, increases (and not decreases) ciliary length (*Figure 2A–C* and *Figure 3*), consistent with the Cdc20 knockdown phenotype. There is no precedent for Cdc20 functioning as an inhibitor of Cdh1 in mitosis. Instead, APC-Cdh1 ubiquitylates Cdc20, leading to its rapid disappearance during anaphase (*Pfleger and Kirschner, 2000*). Finally, we showed that dishevelled is not an authentic substrate of APC-Cdh1 using both specific ubiquitylation assays and degradation assays in vitro (*Figure 5—figure supplement 1C and 1D*). Thus, our data argue against their model and instead point to the role of APC-Cdc20 in negatively regulating ciliogenesis.

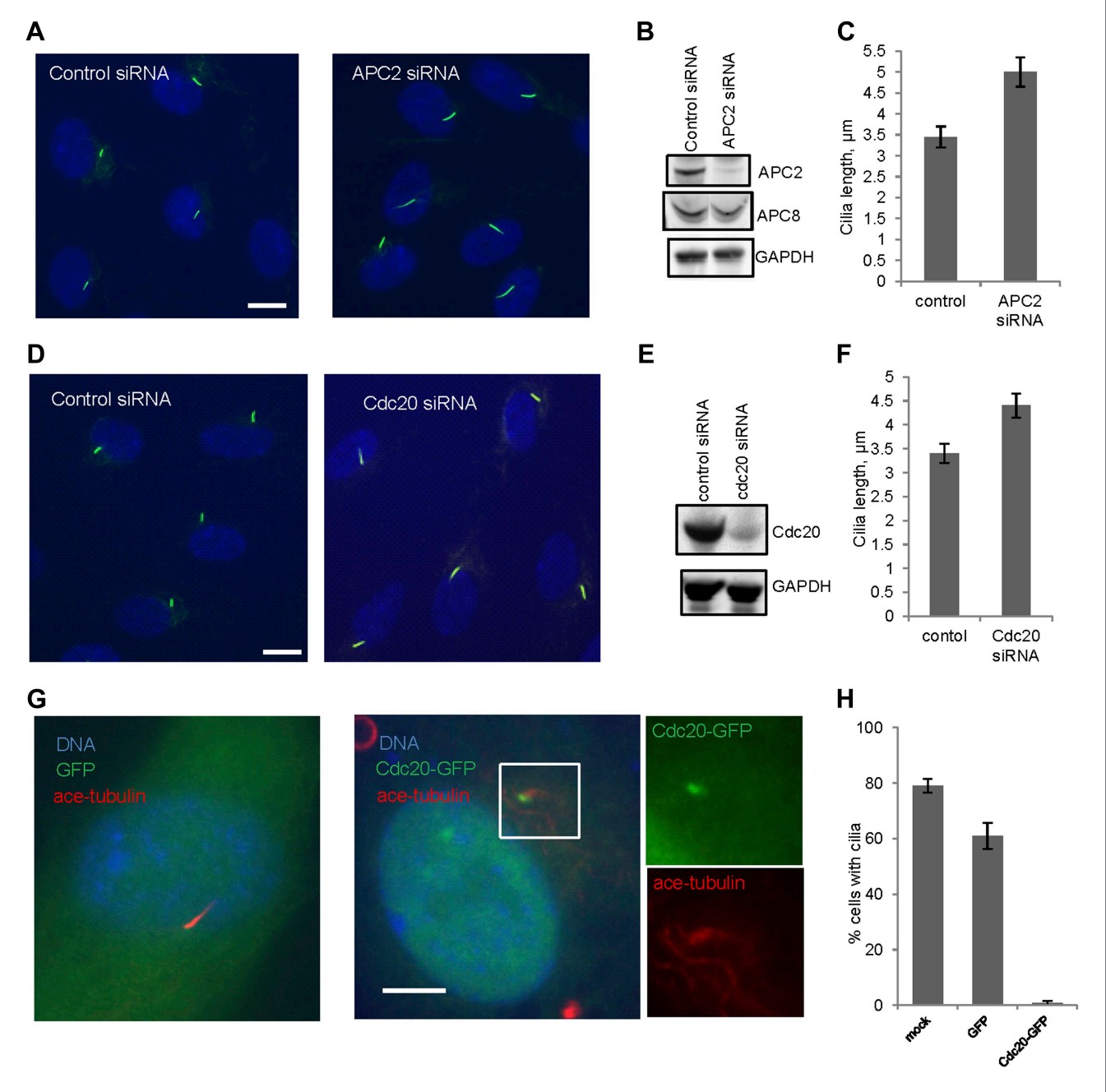

**Figure 2**. APC-Cdc20 negatively regulates the length of primary cilia. (**A** and **D**) RPE1 cells were treated with control siRNA or siRNA targeting APC2 or Cdc20 respectively, serum starved for 48 hr, and stained for acetylated tubulin (green) and DNA (blue). The scale bar represents 10 μm. (**B** and **E**) Western blot showed knockdown efficiency of siRNA treatment against Cdc20 and APc2. (**C** and **F**) Ciliary length was measured from ciliated cells based on acetylated-tubulin staining (n > 30). Data are means ±S.D. p <0.005. (**G**) RPE1 cells were transiently transfected with vectors expressing GFP (green) or Cdc20-GFP (green) respectively, serum starved for 48 hr, and were stained for acetylated tubulin (red) and DNA (blue). The scale bar represents 5 μm. (**H**) Percentage of ciliated cells was obtained from five independent experiments. An average of 100 cells was counted in each of the experiments. Data are means ±S.D.

The following figure supplements are available for figure 2:

**Figure supplement 1**. The cells with longer cilia after inhibition of APC-Cdc20 are still in the quiescent stage.

**Figure supplement 2**. The effects of deregulation of APC co-activators on ciliogenesis.

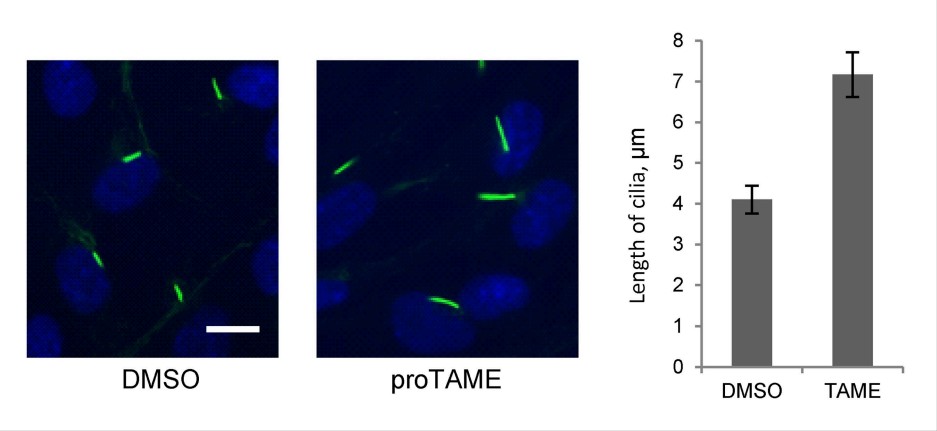

**Figure 3**. Treatment of ciliated cells with proTAME increased the ciliary length. RPE1 cells were serum-starved for 60 hr and further starved for 3 hr in the presence of DMSO or 20 μM proTAME before staining for acetylated-tubulin (green) and DNA (blue). Ciliary length was measured from ciliated cells based on acetylated-tubulin staining (n > 30). Data are means ±S.D. p <0.001. The scale bar: 10 μm.

Furthermore, our data show that APC-Cdc20 counteracts the Nek1 activity to regulate the stability of the ciliary axoneme during maintenance and resorption of cilia. Depletion of Nek1 resulted in branched axonemal microtubules. An explanation for this phenotype is that the function of Nek1 is to stabilize axonemal microtubules. Thus, depletion of Nek1 leads to the destabilization and disassembly of axonemal subunits. It would appear that the disassembled axonemal subunits cannot be transported out of cilia efficiently, and instead they are randomly reassembled in the ciliary lumen, resulting in branched and frayed structures. Depletion of Nek1 also results in defects in ciliary signaling, revealed by the absence of Arl13b staining (*Figure 5—figure supplement 2A*). This indicates that proper regulation of APC-Cdc20 activity is also essential for normal cellular signaling through primary cilia.

We note that APC may control the degradation of a series of ciliary proteins besides Nek1, as this E3 ligase does during mitosis. It will be interesting to try to identify other known and unknown APC substrates in the primary cilium and determine how they are regulated during the process of elongation and resorption.

The regulation of APC activity during mitosis and meiosis is controlled by the activity of kinases, such as Cdk1 (*Kramer at al., 2000*; *Madgwick et al., 2006*). Our preliminary data showed that suppression of Cdk activity increased the ciliary length (unpublished data), consistent with the role of Cdk1 in stimulating APC activity as in mitosis/meiosis. It is also possible that some protein inhibitor of APC is involved in ciliogenesis, regulating the suppression and activation of APC activity during ciliary maintenance and disassembly respectively. It is worthwhile to further dissect the detailed mechanism underlying this process in the future study.

The regulation of the primary cilium has assumed a great importance in overall cellular regulation and disease. The assembly/disassembly of primary cilium is under strict homeostatic control, and it is responsive to cell cycle regulation. The regulatory apparatus bears great similarities to the posttranslational control in the mitotic cycle. Despite the known presence of many of the same components that were thought to be quintessential components of the mitotic engine, it was not clear that their function bears strong similarities to their use in the cell cycle. Central to mitosis is this is the anaphase-promoting complex and its metaphase/anaphase regulator Cdc20. It acts as a master regulator of downstream kinases and phosphatases but it also targets terminal protein substrates. The primary cilium is thought to be a concentrated signaling system whose length may therefore act as a rheostat controlling a complement of signaling activities in cells. It therefore becomes attractive to think of ways of perturbing ciliary length. As more and more evidence accumulates documenting the connection between defects in ciliogenesis and human diseases, there could be a growing interest need for new therapies that modulate ciliary length and ciliary disassembly. An understanding of the central role of the APC in control of ciliary length could aid in the identification of new drug targets that may have broad applicability.

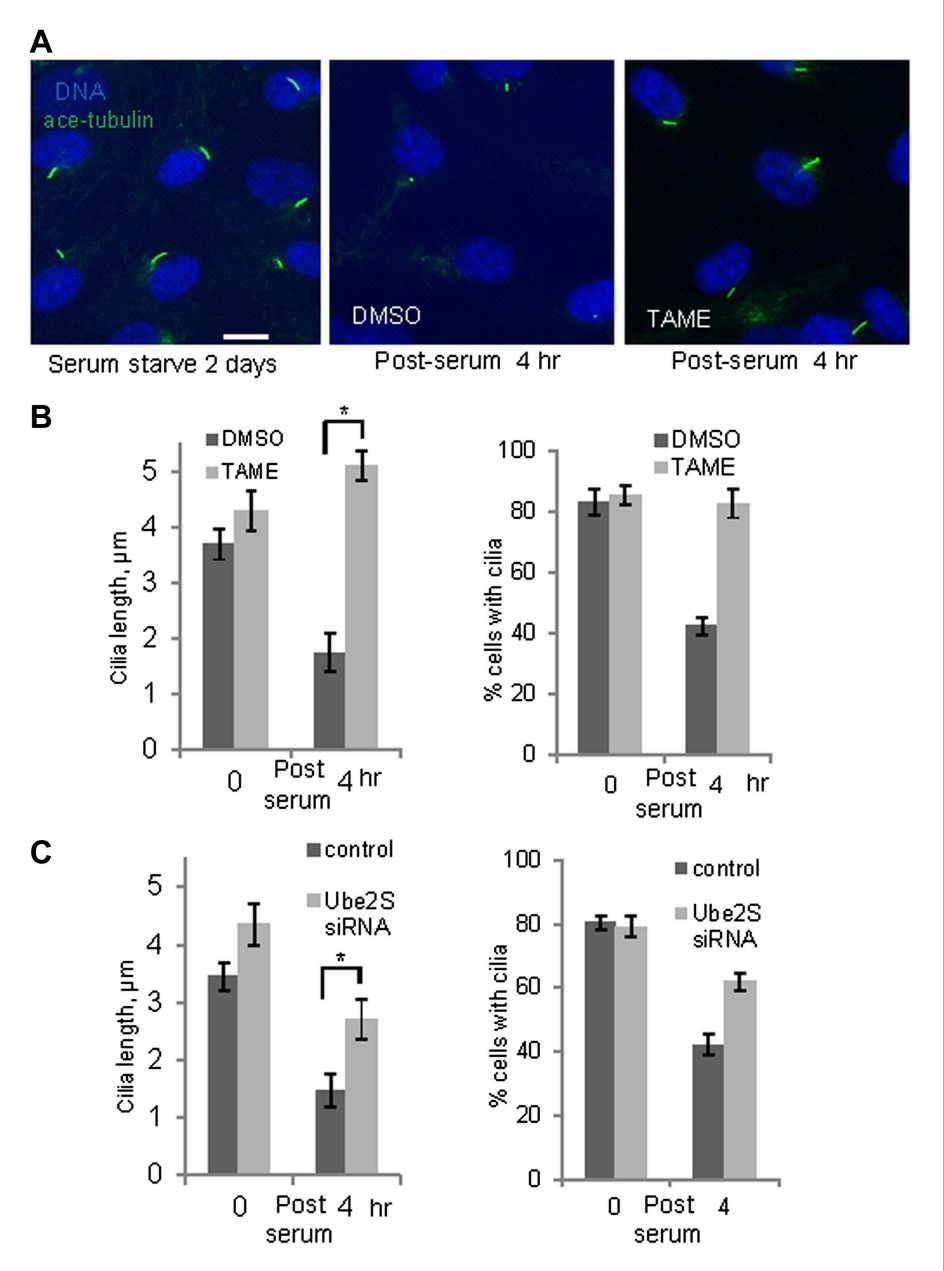

**Figure 4**. Inhibition of APC activity prevented proper ciliary resorption post serum stimulation. (**A**) Quiescent ciliated RPE1 cells were treated with serum containing either DMSO or 20 μM proTAME for 4 hr, fixed and stained for acetylated tubulin (green) and DNA (blue). Scale bar: 10 μm. (**B**) Quantitation of ciliary length and ciliation of DMSO or proTAME treated cells after serum stimulation. Ciliary length was measured from ciliated cells (n > 30) (*p <0.001); the percentage of ciliated cells was obtained from five independent experiments. Data are means ±S.D. (**C**) Cells were treated with control siRNA or Ube2S targeting siRNA and were serum starved for 2 days in the presence of siRNA. Cilia were quantitated after release of cells from starvation for 4 hr. Ciliary length was measured from ciliated cells (n > 30) (*p <0.001); the percentage of ciliated cells was obtained from five independent experiments. Data are means ±S.D.

The following figure supplements are available for figure 4:

**Figure supplement 1**. Suppression of endogenous Cdc20 or APC2 by siRNA delayed cilium resorption.

**Figure supplement 2**. Inhibition of APC activity prevented cilia resorption caused by Calcium influx.

**Figure supplement 3**. RNAi suppression of endogenous Ube2S increased the ciliary length.

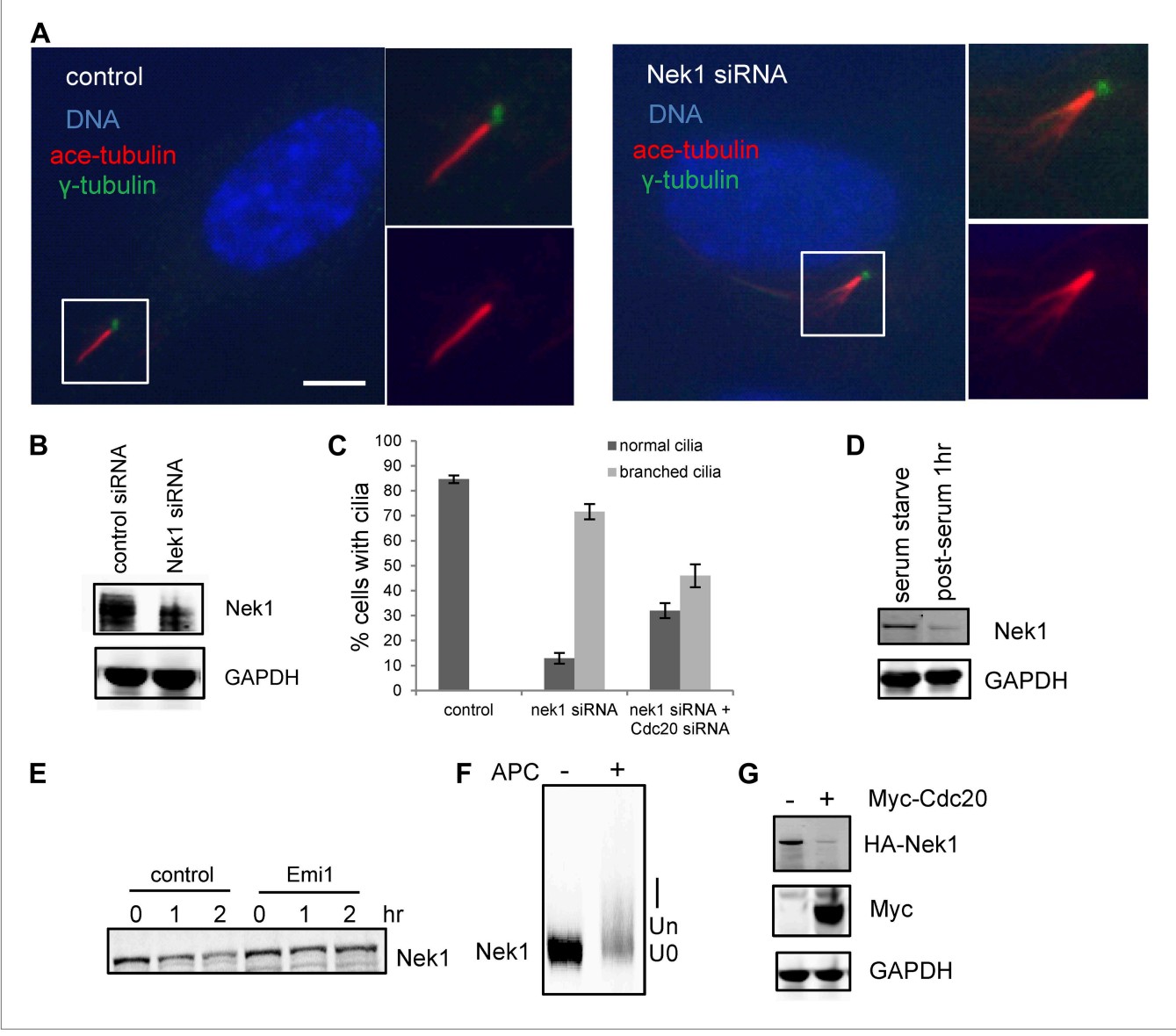

**Figure 5**. Nek1 is the substrate of APC-Cdc20 for the regulation of the structure of primary cilia. (**A**) RPE1 cells were treated with control siRNA or Nek1 targeting siRNA, serum starved for 48 hr, and stained for acetylated tubulin (red), γ-tubulin (green) and DNA (blue). Scale bar represents 5 μm. (**B**) siRNA-treated cells were subjected to Western blot for endogenous Nek1 and GAPDH. (**C**) Cells were transfected with control siRNA, Nek1 siRNA, or Nek1 siRNA together with Cdc20 siRNA respectively, serum starved for 2 days, and stained for acetylated tubulin. The percentage of cells with normal cilia or branched abnormal cilia was calculated from three independent experiments. Data are means ±S.D. (**D**) Western blot showed the decrease of Nek1 protein levels after serum stimulation. (**E**) Degradation of $^{35}$S-labelled Nek1 was examined in cell extracts prepared from HeLa S3 cells. Emi1 was added to test if the degradation was dependent on APC activity. The data were analyzed by autoradiography. (**F**) HA-tagged Nek1 was expressed and labeled with $^{35}$S in reticulocyte lysate. $^{35}$S-labelled Nek1 was purified through HA tag and subjected to in vitro ubiquitylation catalyzed by purified APC. (**G**) RPE1 cells were transfected with plasmid expressing HA-Nek1 and/or Myc-Cdc20, starved for 2 days and subjected to western blot for Myc tag, HA tag, and GAPDH.

The following figure supplements are available for figure 5:

**Figure supplement 1**. Test of ciliary proteins as APC substrates.

**Figure supplement 2**. Depletion of Nek1 caused specific microtubule defects in primary cilia.

# Materials and methods

## Cell culture and transfection

hTERT-RPE1 cells were grown in Dulbecco's Modified Eagle Medium (DMEM) with 10% fetal bovine serum (FBS). For analysis of ciliary assembly, cells were plated at subconfluence on glass coverslips and starved for 48 hr in DMEM without serum to induce cilia formation. To induce ciliary resorption, starved ciliated cells at subconfluence were treated with DMEM medium with 10% serum for 1–4 hr. Plasmids were transiently transfected into cells with FuGENE 6 Transfection Reagent (Promega, Madison, WI). For siRNA treatment, cells were transfected with indicated siRNA by Lipofectamine RNAiMAX Transfection Reagent (Life Technologies, Beverly, MA).

## Plasmids, siRNA, and drugs

Plasmids from the laboratory stock are as follows: human securin, human HA-Nek1, GFP, human Cdc20-GFP, human cMyc-Cdc20, and human DVL in pCS2 vector.

SMARTpool ON-TARGETplus siRNA for Nek1 and Cdc20 were obtained from Thermo scientific, Waltham, MA. Silencer Select siRNA for Ube2S were obtained from Life Technologies.

Calcium ionophore was from Sigma, St. Louis, MO. proTAME was from Boston Biochem, Cambridge, MA.

## Protein preparation, in vitro degradation and ubiquitylation assays

Recombinant Emi1 and APC components are purified as previously described (*Wang and Kirschner, 2013*). Securin and HA-Nek1 were expressed and radiolabelled with $^{35}$S in reticulocyte lysate (Promega). HA-Nek1 was further purified through HA beads (Sigma) for ubiquitylation assay. Cell extracts were prepared from HeLa S3 cells and RPE1 cells as indicated. In vitro protein degradation and in vitro ubiquitylation assays were performed as described previously (*Wang and Kirschner, 2013*).

## Western blot

hTERT-RPE1 cells were lysed in SDS sample buffer, and proteins were separated by SDS-PAGE on 4–12% gel. Data were analyzed with the Odyssey Infra-red imaging system. Primary antibodies for western blot were as follows: anti-GAPDH (1:20,000; 0411; sc47724; Santa Cruz Biotech, Santa Cruz, CA), anti-Nek1 (1:100; PAB3282; Abnova, Taiwan), anti-c-Myc (1:500; 9E10; sc40; Santa Cruz Biotech), anti-HA (1:500; Y11; sc805; Santa Cruz Biotech), anti-Ube2S (1:1000; 2257; Strategic Diagnostics, Newark, DE), anti-actin (1:2000; A2066; Sigma), anti-Aurora A (1:1000; 3092; Cell Signaling, Beverly, MA), anti-Cdc20 (1:1000; AR12; K0140-3; MBL International, Woburn, MA). Infrared secondary antibodies (1:20,000; Li-COR Biosciences, Lincoln, NE) were used for Odyssey imaging.

## Immunofluorescence

Cells on coverslips were fixed with 3.5% paraformaldehyde (10 min) at 37°C and then methanol (4 min) at 4°C, permeabilized with 0.2% Triton-X100 in PBS, blocked in 2.5% BSA in PBS, and incubated with antibodies. Alternatively, to optimize signals at centrosomes or basal bodies, cells were fixed in methanol at −20°C for 10 min and subjected to blocking and antibody incubation. Primary antibodies included anti-APC2 (1:50; home-made), anti-APC7 (1:50; 21,418; Santa Cruz Biotech), anti-Cdc20 (1:100; AR12; K0140-3; MBL International), anti-γ-tubulin (1:5000; T5192; Sigma), anti-acetylated tubulin (K40) (1:2000; 6-11B-1; T7451; Sigma), anti-acetylated tubulin (K40) (1:2000; D20G3; 5335; Cell Signaling), anti-p150Glued (1:100; 1/p150Glued; 610473; BD Biosciences, San Jose, CA), anti-Arl13b (1:500; 17,711-1-AP; Proteintech, Chicago, IL), anti-ki-67 (1:500; ab15580; Abcam, Cambridge, MA), anti-centrin 1 (1:200; ab11257; Abcam), anti-Cdh1 (1:100; home-made), anti-Cdh1 (1:200; Ab-2; Calbiochem), anti-Cdh1 (1:200; 34-2000; Life Technologies). Secondary antibodies labeled with Alexa 488 and Alexa 568 were from Life Technologies. DNA was stained with DAPI. Images were acquired with Nikon 80i Upright Microscope, and processed with MetaMorph image acquisition software and Adobe Photoshop SC3.

In order to measure the length of cilia, during coverslip mounting, the coverslips were pressed hard against microscope slides to flatten most of the cilia. Only those cilia which were flattened into one plane were selected for image acquirement and length measurement.

# Acknowledgements

We wish to thank Dr Jagesh V Shah at Harvard Medical School for discussions and insightful suggestions. We thank Margaret Coughlin at Harvard Medical School for her effort to confirm the cilium phenotype with electron microscopy. We are grateful to the Nikon imaging center at Harvard Medical School and

in particular to Jennifer Waters for technical support. The work was supported by a grant from the National Institute of General Medical Sciences, R01 GM039023.

## Additional information

### Funding

| Funder | Grant reference number | Author |
| --- | --- | --- |
| National Institute of General Medical Sciences | R01 GM039023 | Marc W Kirschner |

The funder had no role in study design, data collection and interpretation, or the decision to submit the work for publication.

### Author contributions

WW, Conception and design, Acquisition of data, Analysis and interpretation of data, Drafting or revising the article; TW, Conception and design, Acquisition of data; MWK, Analysis and interpretation of data, Drafting or revising the article

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
