## [Decision Letter]

Thank you for sending your work entitled “APC-Cdc20 regulates ciliary length and disassembly of the primary cilium” for consideration at *eLife*. Your article has been favorably evaluated by Randy Schekman (Senior editor) and 3 reviewers, one of whom is a member of our Board of Reviewing Editors.

The Reviewing editor and the other reviewers discussed their comments before we reached this decision, and the Reviewing editor has assembled the following comments to help you prepare a revised submission.

Your study focuses on the regulation of primary cilium length, and reabsorption. This is a poorly understood area of cilium biology, and this work provides novel and interesting insights into a mechanism involving the anaphase-promoting complex APC/Cdc20. You confirm a previous study that APC/Cdc20 localize to the ciliary basal body. Using specific inhibitors, siRNAs and over-expression they then show that the APC/Cdc20 complex is required for controlling cilium length, and reabsorption upon serum-induced re-entry into the cell cycle. Significantly you show that the ciliary kinase Nek1 is an APC/Cdc20 substrate. Furthermore, the study addresses some inconsistencies in previous studies.

Despite these overall positive comments, the reviewers agreed that there are flaws in your study that need to be addressed before it is considered further.

General comments:

1) It is clear from the Discussion that the work is in contradiction to a prior study addressing the roles of APC, Cdc20, and Cdh1. However, because these issues are not raised until the Discussion, the presentation of data is not structured to rigorously allow critique of the prior work: many additional controls, and performance of side-by-side comparison of Cdc20 with Cdh1 throughout the work, would be necessary if this is the main point to be made. For example, it is stated that Cdh1 is not localized at the basal body, contrary to a previous study. However, the experimental conditions and antibodies used in your study are not discussed (were different antibodies used?), nor is there discussion of disheveled or its relevance to cilia in the main text. For example, it would be important to show parallel data for all experiments with Cdh1, rather than stating as unpublished data that the knockdown of this protein gave no ciliary effect. These data should include evidence that efficient knockdown was achieved. If you do not wish to do these experiments, then the discussion should be modified.

2) Some of the data do not support the interpretation of observed results, particularly with Nek1, and some simple but important control experiments are missing. For example, if the action of Cdc20 is to promote Nek1 degradation, then overexpression of Nek1 or Nek1 with a mutated recognition sequence for APC should lengthen cilia under steady state conditions, or prevent ciliary resorption following induction of disassembly. This experiment would test your model directly.

3) It is not clear how APC/Cdc20 activity is regulated – of particular importance is the finding that the level of at least Cdc20 is high at 48 hr serum starvation (Figure 1—figure supplement 2), but cilia are still present; is proteolysis of Nek1 inhibited? What happens upon serum add-back and cilium reabsorption? Are levels of APC/Cdc20 decreased, or are APC/Cdc20 relocalized from the basal body?

Specific comments:

1) Is the APC present at the centriole(s) of a quiescent cell that does not have a cilium? This is important because some components of the centrosome/cilium complex only localize to that complex after the cilium compartment is formed (the transition zone components, for example), and it would be useful to know the APC's relationship to other components in the pathway of ciliogenesis. Similarly, in Figure 1, it is stated that Cdc20 is not observed at the centrosome of cycling cells. However, it I important to show that there is no accompanying axoneme (e.g., stained by acetylated tubulin). This would be necessary to make the claim that staining is centrosomal rather than at the ciliary basal body, as many cilia persist in non-mitotic interphase cells.

2) What criteria are used to conclude that the “branched axoneme” shown in Figure 5 under conditions of Nek1 depletion has the structure of a ciliary axonome, such as doublet microtubules, and is surrounded by a ciliary membrane? It looks like an internal microtubule array associated with the centriole, which could be an aberrant axoneme, but might be something else. The lack of Arl13b, cited by the authors as evidence that there is a signaling defect under these circumstances, might actually be due to this not being a ciliary axoneme.

3) Figure 1—figure supplement 1: APC is still active in quiescent extracts based on an assay performed with degradation of securin. This assay should be used in parallel with lysates depleted for Cdc20, to make it clear that Cdc20 is important for this function; and with lysates treated with pro-TAME, as a control. As presented, this assay lacks a negative control. Figure 1—figure supplement 2: Colocalization with the dynactin subunit P150 is used to demonstrate that the APC is associated with the mother centriole. This is not clear from the images in the Figure, because there is no marker of all centrioles. Ideally this experiment would be done with triple labeling for APC2, P150 and a marker such as centrin.

4) Figure 2: This figure shows the change in cilium length associated with compromising APC-Cdc20. There is no indication of how the length of cilia was measured, and this is important because the length is used throughout the paper and in some cases the change is modest in amount. To determine cilium length accurately, the z-axis must be taken into account.

5) Figure 4: According to the legend, “Cilia were visualized by acetylated tubulin staining (green)”. The legend should also mention the DAPI staining of DNA in all of the panels. More importantly, it is not clear what is shown in the ionophore panel. What is the yellow labeling, and why is it shown as yellow if the intention was to show the acetylated-tubulin in green? Is the signal from the normal complement of stable acetylated microtubules?

6) It is unclear why data on Aurora-A kinase are not shown, even though this kinase is known to be a proximal regulator of ciliary assembly that is localized to the basal body and targeted by the APC. The authors exclude Aurora-A by stating it is phosphorylated at S51 during ciliary resorption, protecting it from APC-mediated degradation. However, the cited reference does not indicate this information is contained in the paper. In general, the weakness of the data presented for Nek1 makes it important to examine at least minimally the interaction between Cdc20 and Aurora-A in this study. The authors could test this more directly by show effects of depletion or over-expression of Cdc20 on Aurora-A expression and localization to the basal body.

---

## [Author Response]

General comments:

*1) It is clear from the Discussion that the work is in contradiction to a prior study addressing the roles of APC, Cdc20, and Cdh1. However, because these issues are not raised until the Discussion, the presentation of data is not structured to rigorously allow critique of the prior work: many additional controls, and performance of side-by-side comparison of Cdc20 with Cdh1 throughout the work, would be necessary if this is the main point to be made. For example, it is stated that Cdh1 is not localized at the basal body, contrary to a previous study. However, the experimental conditions and antibodies used in your study are not discussed (were different antibodies used?), nor is there discussion of disheveled or its relevance to cilia in the main text. For example, it would be important to show parallel data for all experiments with Cdh1, rather than stating as unpublished data that the knockdown of this protein gave no ciliary effect. These data should include evidence that efficient knockdown was achieved. If you do not wish to do these experiments, then the discussion should be modified*.

a) As suggested by reviewers, the Cdh1 knockdown data including western blot to show knockdown efficiency and calculation of ciliary length were added (Figure 2—figure supplement 2). The result shows that Cdh1 knockdown did not significantly affect the assembly of cilia, contradictory to the model by *Miyamoto et al.* proposing that APC-Cdh1 promotes ciliogenesis (17). The work by Miyamoto et al. did not report any effect of knockdown of Cdh1 on ciliogenesis.

b) We tried three different antibodies against Cdh1 for immunofluorescence assay (Figure 1—figure supplement 3), and none of them shows its localization to the basal body. The work by Miyamoto et al. did not report an immunofluorescence assay to show the cellular localization of Cdh1, which makes it hard for thorough comparison with our Cdc20 data.

c) As suggested, we modified the data presentation in Results section for comparison of our work to the previous study. We also moved the data about dishevelled and Aurora-A to the main text (Results section), along with interpretation of the results.

*2) Some of the data do not support the interpretation of observed results, particularly with Nek1, and some simple but important control experiments are missing. For example, if the action of Cdc20 is to promote Nek1 degradation, then overexpression of Nek1 or Nek1 with a mutated recognition sequence for APC should lengthen cilia under steady state conditions, or prevent ciliary resorption following induction of disassembly. This experiment would test your model directly*.

We appreciate this suggestion, and actually we tested the effect of overexpression of Nek1. The problem is that overexpression of Nek1 disrupted the overall cytosolic microtubule structure, leading to the total absence of ciliary structure, consistent with a previous study (28). Thus, it is hard to study the effect of Nek1 on cilia with overexpression, although we did find that overexpression of a kinase dead version of nek1 also caused branched ciliary structure presumably due to dominant negative effect.

*3) It is not clear how APC/Cdc20 activity is regulated – of particular importance is the finding that the level of at least Cdc20 is high at 48 hr serum starvation (*Figure 1—figure supplement 2*), but cilia are still present; is proteolysis of Nek1 inhibited? What happens upon serum add-back and cilium reabsorption? Are levels of APC/Cdc20 decreased, or are APC/Cdc20 relocalized from the basal body?*

a) We repeated the western blot in the new Figure 1—figure supplement 2 (Figure 1—figure supplement 2 in the previous version), and added time points post serum addition. It shows that the protein levels of Cdc20 did not change after serum addition. We also did the immunofluorescence assay, and showed that Cdc20 is still at the basal body during cilium resorption (Figure 1 middle panel).

b) We suspect that some APC inhibitor protein might be involved in partial inhibition of APC-Cdc20 during cilium maintenance and fully activation of APC during ciliary resorption, as APC inhibitors act during both mitosis and meiosis. Indeed, our preliminary data showed that GFP-conjugated Emi2 was concentrated to the basal body of primary cilium, and transfection of cells with siRNA against Emi2 may reduce the ciliary length. However, unavailability of a good antibody against endogenous Emi2 for either western blot or immunofluorescence assay limits further investigation of Emi2 function during ciliogenesis.

Specific comments:

*1) Is the APC present at the centriole(s) of a quiescent cell that does not have a cilium? This is important because some components of the centrosome/cilium complex only localize to that complex after the cilium compartment is formed (the transition zone components, for example), and it would be useful to know the APC's relationship to other components in the pathway of ciliogenesis. Similarly, in*
Figure 1*, it is stated that Cdc20 is not observed at the centrosome of cycling cells. However, it I important to show that there is no accompanying axoneme (e.g., stained by acetylated tubulin). This would be necessary to make the claim that staining is centrosomal rather than at the ciliary basal body, as many cilia persist in non-mitotic interphase cells*.

We did the corresponding experiment to answer these comments. Interphase proliferation cells were co-staining for Cdc20 and acetylated tubulin as requested. As shown in Figure 1 (lower panel), Cdc20 is not localized to centrosome in non-ciliated interphase cells. In addition, we showed that APC2 is not concentrated to centrosome in non-ciliated quiescent cells (Figure 1—figure supplement 2). Thus, we concluded that APC-Cdc20 is only recruited to the basal body after cilium is formed.

*2) What criteria are used to conclude that the “branched axoneme” shown in*
Figure 5
*under conditions of Nek1 depletion has the structure of a ciliary axonome, such as doublet microtubules, and is surrounded by a ciliary membrane? It looks like an internal microtubule array associated with the centriole, which could be an aberrant axoneme, but might be something else. The lack of Arl13b, cited by the authors as evidence that there is a signaling defect under these circumstances, might actually be due to this not being a ciliary axoneme*.

There are several evidence indicating that the branched structures are defective cilia. First, the branched structure does not appear in non-ciliated proliferating cells after nek1 knockdown (Figure 5—figure supplement 2). Second, the assembly and disassembly of the branched structure follow the same dynamics as normal cilia: It assembles after starvation (Figure 5), and resorbs after serum stimulation (Figure 5—figure supplement 2).

*3)*
Figure 1—figure supplement 1*: APC is still active in quiescent extracts based on an assay performed with degradation of securin. This assay should be used in parallel with lysates depleted for Cdc20, to make it clear that Cdc20 is important for this function; and with lysates treated with pro-TAME, as a control. As presented, this assay lacks a negative control.*
Figure 1—figure supplement 2*: Colocalization with the dynactin subunit P150 is used to demonstrate that the APC is associated with the mother centriole. This is not clear from the images in the Figure, because there is no marker of all centrioles. Ideally this experiment would be done with triple labeling for APC2, P150 and a marker such as centrin*.

a) As requested by reviewers, we repeated the degradation assay in Figure 1—figure supplement 1, and added controls including the assay after Cdc20 depletion and the assay treated with proTAME respectively.

b) To address the second comment, we co-stained quiescent RPE1 cells for P150 and centrin 1, and showed that P150 is localized to one centrin spot (Figure 1—figure supplement 2), which is consistent with a previous report (8) showing that P150 co-localizes with mother centriole. This, together with the co-localization of APC2 with P150 (Figure 1—figure supplement 2), support the conclusion that APC is localized to the mother centriole.

*4)*
Figure 2*: This figure shows the change in cilium length associated with compromising APC-Cdc20. There is no indication of how the length of cilia was measured, and this is important because the length is used throughout the paper and in some cases the change is modest in amount. To determine cilium length accurately, the z-axis must be taken into account*.

In order to measure the length of cilia, during coverslip mounting, the coverslips were pressed hard against microscope slides to flatten most of the cilia. Only those cilia which were flattened into one plane were selected for image acquisition and length measurement. We added this description of length measurement in the Methods section.

*5)*
Figure 4*: According to the legend, “Cilia were visualized by acetylated tubulin staining (green)”. The legend should also mention the DAPI staining of DNA in all of the panels. More importantly, it is not clear what is shown in the ionophore panel. What is the yellow labeling, and why is it shown as yellow if the intention was to show the acetylated-tubulin in green? Is the signal from the normal complement of stable acetylated microtubules?*

As suggested by reviewers, we have revised the figure legend (the description of DNA staining was added), and corrected the error in image processing in Figure 4—figure supplement 2 (in the previous version of the figure, we put the acetylated tubulin staining into both red and blue channel by mistake, causing the yellow color).

*6) It is unclear why data on Aurora-A kinase are not shown, even though this kinase is known to be a proximal regulator of ciliary assembly that is localized to the basal body and targeted by the APC. The authors exclude Aurora-A by stating it is phosphorylated at S51 during ciliary resorption, protecting it from APC-mediated degradation. However, the cited reference does not indicate this information is contained in the paper. In general, the weakness of the data presented for Nek1 makes it important to examine at least minimally the interaction between Cdc20 and Aurora-A in this study. The authors could test this more directly by show effects of depletion or over-expression of Cdc20 on Aurora-A expression and localization to the basal body*.

a) We added additional experiment showing that the protein levels of Aurora-A did not significantly change after serum stimulation (Figure 5—figure supplement 1), consistent with the result in the original paper revealing Aurora-A function in cilia (25).

b) As requested, we also did siRNA transfection and western blot, and found that Cdc20 depletion did not significantly affect Aurora-A protein levels during ciliogenesis (Figure 5—figure supplement 1).

c) We added the references indicating that Aurora-A is phosphorylated at S51 during ciliogenesis (24), and that phosphorylation protects Aurora-A from APC-mediated degradation during mitosis (13; 14). We also put the discussion about Aurora-A in the Results section.